# Monitoring Mining Surface Subsidence with Multi-Temporal Three-Dimensional Unmanned Aerial Vehicle Point Cloud

Xiaoyu Liu [1], Wu Zhu [1,2,*], Xugang Lian [3] and Xuanyu Xu [1]

1. School of Geological Engineering and Geomatics, Chang'an University, Xi'an 710054, China
2. Key Laboratory of Ecological Geology and Disaster Prevention, Ministry of Natural Resources, Xi'an 710054, China
3. School of Mining Engineering, Taiyuan University of Technology, Taiyuan 030024, China
* Correspondence: zhuwu@chd.edu.cn

**Abstract:** Long-term and high-intensity coal mining has led to the increasingly serious surface subsidence and environmental problems. Surface subsidence monitoring plays an important role in protecting the ecological environment of the mining area and the sustainable development of modern coal mines. The development of surveying technology has promoted the acquisition of high-resolution terrain data. The combination of an unmanned aerial vehicle (UAV) point cloud and the structure from motion (SfM) method has shown the potential of collecting multi-temporal high-resolution terrain data in complex or inaccessible environments. The difference of the DEM (DoD) is the main method to obtain the surface subsidence in mining areas. However, the obtained digital elevation model (DEM) needs to interpolate the point cloud into the grid, and this process may introduce errors in complex natural topographic environments. Therefore, a complete three-dimensional change analysis is required to quantify the surface change in complex natural terrain. In this study, we propose a quantitative analysis method of ground subsidence based on three-dimensional point cloud. Firstly, the Monte Carlo simulation statistical analysis was adopted to indirectly evaluate the performance of direct georeferencing photogrammetric products. After that, the operation of co-registration was carried out to register the multi-temporal UAV dense matching point cloud. Finally, the model-to-model cloud comparison (M3C2) algorithm was used to quantify the surface change and reveal the spatio-temporal characteristics of surface subsidence. In order to evaluate the proposed method, four periods of multi-temporal UAV photogrammetric data and a period of airborne LiDAR point cloud data were collected in the Yangquan mining area, China, from 2020 to 2022. The 3D precision map of a sparse point cloud generated by Monte Carlo simulation shows that the average precision in X, Y and Z directions is 44.80 mm, 45.22 and 63.60 mm, respectively. The standard deviation range of the M3C2 distance calculated by multi-temporal data in the stable area is 0.13–0.19, indicating the consistency of multi-temporal photogrammetric data of UAV. Compared with DoD, the dynamic moving basin obtained by the M3C2 algorithm based on the 3D point cloud obtained more real surface deformation distribution. This method has high potential in monitoring terrain change in remote areas, and can provide a reference for monitoring similar objects such as landslides.

**Keywords:** unmanned aerial vehicle (UAV); structure-from-motion (SfM); airborne LiDAR; precision maps; co-registration; point cloud; dynamic moving basin

## 1. Introduction

As a main energy source in the world, coal plays an important role in national economic construction [1]. However, the activities of intensive mining may cause severe surface subsidence, and subsequently damage buildings, infrastructure, roads and the ecological environment [2,3]. It is necessary to monitor the surface subsidence in mining areas with purpose of mitigating these damages. Traditional ground-based monitoring methods,

such as total station and leveling, can provide high-precision measurements at the level of centimeters to millimeters. The problems of ground-based methods are the high cost, point-like observation and the difficulty in monitoring remote areas, limiting the further uses of these tools [4,5].

The development of remote sensing technologies, such as interferometric synthetic aperture radar (InSAR) and unmanned aerial vehicles (UAV), has promoted the acquisition of high-resolution terrain data and therefore improved the ability to monitor terrain changes with wide coverage and high accuracy [6,7]. InSAR has shown its potential in monitoring coal mining subsidence due to the advantages of all-weather, all-time operation, and an accuracy of centimeters to millimeters [8–10]. The main limitation of InSAR technology in coal mining is the decorrelation since mining-induced deformation always exceeds the InSAR maximum detectable deformation.

With the development of the UAV platform and computer vision technology, more and more camera-captured images are processed by using structure from motion (SfM) and multi-view stereo (MVS) algorithms to produce high-resolution terrain data, namely, point clouds, digital surface models (DSM) and digital orthophoto maps (DOM) [11]. These products are further used in the field of precision agriculture [12,13], landslide monitoring [14,15], coastal decline [16,17] and glacier dynamics [18,19]. Compared with the InSAR or airborne LiDAR, the advantages of UAV SfM photogrammetry are flexibility, suitability for complex terrains, and the option for users to define the spatio-temporal resolution to generate time-series terrain changes. These characteristics promote the application of UAV in coal mining subsidence monitoring [20].

The most common method of UAV subsidence monitoring is calculating the vertical or horizontal displacement by differencing two images which are gridded through interpolating point clouds [21–26]. This process may introduce errors in complex natural topographic environments. Therefore, a complete three-dimensional change analysis is required to quantify the surface change in complex natural terrain. The method based on point clouds can capture the multi-directional surface changes and accurately monitor the spatial displacement characteristics, which has been successfully used in rock glaciers [27,28], landslides [29,30], and other fields. There are relatively few studies on mining subsidence monitoring using the point cloud method. In addition, the reliability of multi-temporal point cloud change analysis on complex terrain surfaces is affected by the uncertainty sources such as point density, surface roughness and registration error.

Georeferencing is the key to obtaining repeatable data to monitor surface changes. With the development of global navigation satellite system (GNSS) technology and accurate time measuring systems, UAV equipped with real time kinematic (RTK) and post processed kinematic (PPK) can accurately determine the position and orientation of UAV at the imaging moment. UAV photogrammetry has made a breakthrough in reducing the number of ground control points (GCPs), and even not using GCPs [31,32]. Considering the rapid development of UAV technology, it is necessary to evaluate the accuracy of UAV georeferencing results obtained by the GNSS RTK/PKK technology. Nolan et al. obtained the data with a ground sampling distance (GSD) of 10–20 cm in an area of dozens of square kilometers, and verified that the accuracy and precision (repeatability) of direct georeferencing are better than ±30 and ±8 cm, respectively [33]. Padr et al. evaluated the UAV georeferencing accuracy of a farm, and showed that the horizontal and vertical RMSE were not greater than 0.256 and 0.238 m, respectively [34]. Precision assessment benefits from a larger and more evenly distributed number of checkpoints, which are often difficult in areas with undulating terrain and steep slopes where it is difficult for people to access. Therefore, it is necessary to evaluate the precision of photogrammetric products without ground control.

The purpose of this study is to evaluate the relative precision of photogrammetric products without ground control, and then to quantify the topographic changes caused by mining subsidence. Four UAV photogrammetric datasets and one airborne LiDAR point cloud dataset from 2020 to 2022 were collected in the Yangquan mining area, China, to

evaluate the performance of the proposed method. Firstly, the three-dimensional precision of sparse point clouds was obtained through Monte Carlo simulation. Then, the airborne LiDAR point cloud was used as the reference data to co-register the multi-temporal UAV dense matching point cloud. Finally, the surface changes were quantified through the multiscale model-to-model cloud comparison (M3C2) algorithm to reveal the spatio-temporal development characteristics.

## 2. Study Area and Data

### 2.1. Study Area

The study area is a working face of No. 1 Coal Mine of Yangquan Coal Group, Shanxi Province, as shown in Figure 1. The characteristics of the working face are as follows: the strike length is 1345 m, the dip length is 226 m, the average dip angle of the coal seam is 4°, the average mining depth is 446.8 m, and the average coal thickness is 7.24 m. The working face began in October 2019 and was completed in April 2021. The terrain of the mining area is high in the middle, low in the south and north, and the surface is covered with vegetation. A half strike and dip observations were conducted at A and B in the study area, respectively. The black rectangle in the Figure 1 is the boundary of the mining working face.

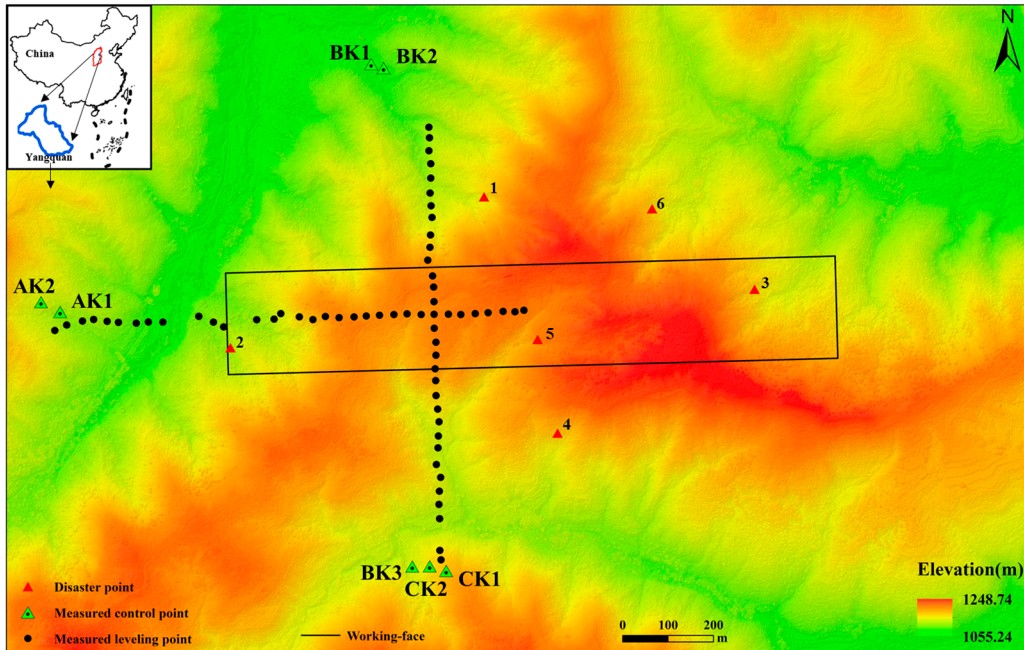

**Figure 1.** Relative position relationship of working face, observation station, and study area.

Figure 2 shows the mining-induced surface changes, and we can see the obvious environmental damages including the destroyed vegetation, landslide, ground fissures and sinkholes. These damages could have different degrees of influence on roads, buildings, and other infrastructures.

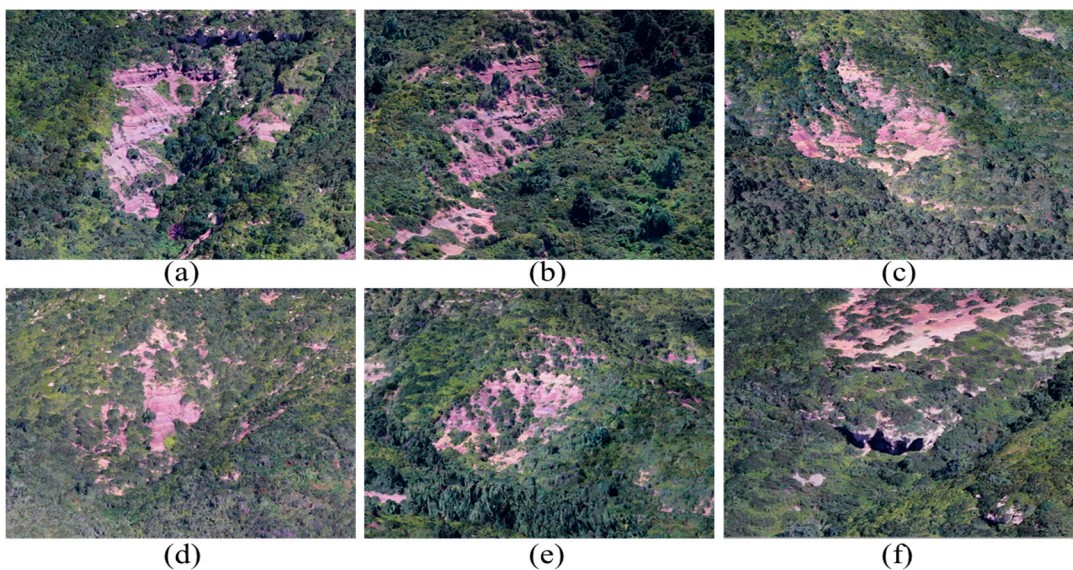

**Figure 2.** Mining-induced surface changes. (**a**–**e**) landslide; (**f**) collapse.

*2.2. Data*

2.2.1. Data Acquisition

Five UAV datasets between March 2020 and January 2022, including one airborne LiDAR dataset and four UAV photogrammetric datasets, were acquired in the study area, as shown in Table 1. The data collected on 14 June 2020 will be abbreviated as 06.14 in the following, and the data in other periods are similar.

**Table 1.** The collected UAV datasets in the study area.

| Number | Time | Acquisition Mode | Achievements Form | Number of Point Clouds | Area km$^2$ | Density per m$^2$ |
|---|---|---|---|---|---|---|
| 1 | 2020.06.14 | UAV image | Point cloud, DSM, DOM | $2.3 \times 10^8$ | 4.5 | 52 |
| 2 | 2020.07.20 | UAV image | Point cloud, DSM, DOM | $2.4 \times 10^8$ | 4.5 | 53 |
| 3 | 2020.09.07 | UAV image | Point cloud, DSM, DOM | $2.2 \times 10^8$ | 4.2 | 53 |
| 4 | 2021.07.31 | UAV image | Point cloud, DSM, DOM | $4.7 \times 10^8$ | 4.3 | 110 |
| 5 | 2022.01.16 | Airborne LiDAR | Point cloud | $2.5 \times 10^8$ | 4.0 | 62 |

The D2000 quad-rotor UAV (Feima Robotics Technology Co., Ltd., Shenzhen, China) was used to carry the D-CAM2000 aerial survey module, as shown in Figure 3a. The platform is equipped with a SONY a6000 digital camera, with a focal length of about 25 mm and an image size of 6000 × 4000 pixels. Compared with the fixed-wing UAV, the D2000 requires only a small take-off and landing space, and has high flexibility and stability in most situations, which is advantageous in mountainous terrain conditions. The main parameters input in the route planning stage are relative flight altitude, GSD and image overlap. GSD of 4 cm/pixel can be provided at a flight altitude of 255 m relative to the ground. An 80% forward overlap and 60% side overlap are set for each flight to ensure the overlap of consecutive photos, which is essential for the image matching algorithm used in SfM.

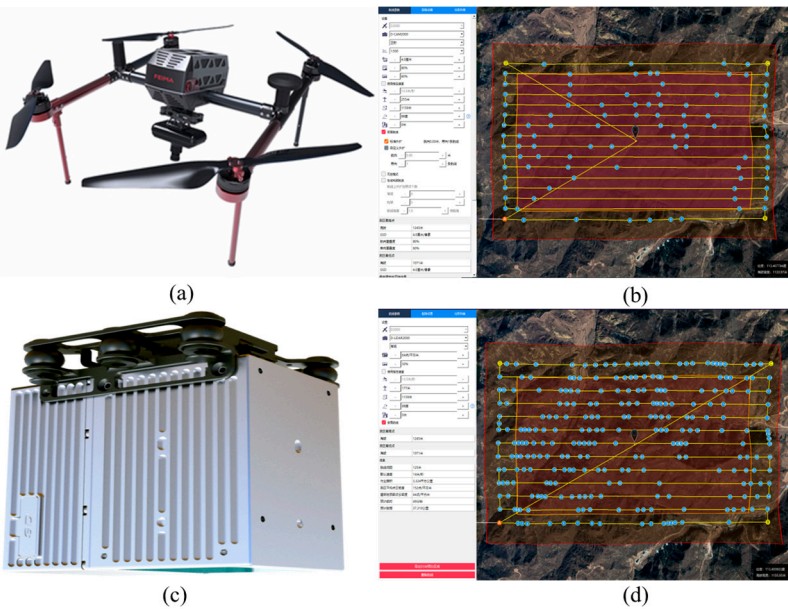

**Figure 3.** UAV platform and route planning. (**a**) Optical lens; (**b**) Schematic diagram of photogrammetric route planning; (**c**) D-LiDAR2000 module; (**d**) Schematic diagram of airborne LiDAR route planning.

The airborne LiDAR data was collected using the Feima D2000 equipped with the D-LiDAR2000 LiDAR module, as shown in Figure 3c. The route was planned according to the site topographic relief and expected point cloud density, and the overlap was set to not less than 50%. The average flight speed was 14 m/s, and the point cloud with a density of 64 points/m$^2$ was obtained at a flight altitude of 177 m relative to the ground. The raw data includes: lvx format laser data, load IMU file, bin format flight log file, fmcompb format airborne GPS observation data, and fmnav format RTK trajectory file. Detailed information about the camera and LiDAR modules is shown in Table 2.

**Table 2.** Parameter configuration of aerial survey module and LiDAR module.

| D-CAM2000 Aerial Module | | D-LiDAR2000 LiDAR Module | |
|---|---|---|---|
| Camera | SONY a6000 | Ranging | 190 m@10%Reflectivity@100 klx 450 m@80%Reflectivity@0 klx |
| Effective pixels | 24.3 million | Scanning frequency | 240 kHz |
| Sensor | 23.5 × 15.6 mm(aps-c) | Ranging accuracy | ±2 cm |
| Focal length | 25 mm | Horizontal positioning accuracy | 0.02 m |

In addition to the multi-temporal data from the UAV, ground-based observations were obtained. Using GPS for connection measurements, with not less than five sets of static GPS receivers for static measurement, the static plane and elevation accuracy are 5 mm + 1 ppm × D and 1 cm + 1 ppm × D, respectively. RTS-332 total station with the nominal angle measurement accuracy of 2″ and ranging accuracy of 3 mm was used for daily observation and comprehensive observation, including GPS RTK measurements for some monitoring points located in complex terrain. The leveling instrument used in the patrol survey was DL-101C, and the round-trip leveling accuracy of 1 km was 3 mm. Based on the accuracy of all instruments, the accuracy of this field observation was controlled at the centimeter level. The control and monitoring points of the observation station were buried by concrete pouring, and a steel bar was used as the material to mark the observation signs. A total of 23 comprehensive observations spanning from July 2019 to November 2021 were obtained, which provided the reference data for calculating the surface movement. Among them, the data collected on 22 June 2020, 21 July 2020, 23 September 2020 and 30 June 2021 were consistent in time with the UAV images.

2.2.2. UAV Data Processing

Direct georeferencing of the UAV images was conducted through the high precision differential GNSS. First, the position data of each photo was solved through the network RTK/PKK fusion differential operation mode. This mode gives priority to the PPK fixed solution results so as to reduce the risk of data loss due to link interruption. For the non-fixed solution part of the PPK, the RTK fixed solution data was used for fusion. The image exchangeable image file format (EXIF) was then written in turn, where GPS positioning data is stored in the image's header file so that the GPS data can be read directly from the image in the imported software.

These geotagged images were processed using Agisoft Photoscan Professional Version 1.7.0 software (https://www.agisoft.com/, accessed on 15 June 2020) [35]. The software uses an SfM algorithm to generate the point cloud, DSM and DOM of the measured area. The process includes four main steps: feature extraction and matching, bundle adjustment, dense matching and generating DSM and DOM. Firstly, the feature points were extracted, which are invariant to the scale and rotation of the image, and also maintain a certain stability to the change of illumination and 3D camera perspective. Based on each pair of overlapping images, the feature matching was performed and the external direction parameters were solved; the model was georeferenced with the additional position information of the image, and the camera position was optimized by minimizing the error between the position of the model points and the measurement position, while reducing the nonlinear deformation; the dense point cloud was built, the point density was increased by several orders of magnitude, and the DSM and DOM were generated on this basis. The original LiDAR data was preprocessed by trajectory solution, point cloud solution, feature extraction and strip adjustment to obtain the point cloud in las format. The working platform for UAV data processing was Dell Tower Workstation (Intel (R) Xeon (R) W-2245 CPU @ 3.90 GHz 3.91 GHz, 128 GB RAM). It took about six hours to complete the workflow of UAV image data processing for each survey.

Point cloud post-processing included point cloud denoising and filtering to remove the non-ground points. The outliers in the original multi-temporal point cloud were removed by using the statistical outlier removal (SOR) tool in CloudCompare software (2.10, https://www.danielgm.net/cc/, accessed on 15 June 2020) [36], as shown in Figure 4. The filter performs a statistical analysis on the neighborhood of each point and calculates its average distance to all nearby points. Assuming that the result is a Gaussian distribution whose shape is determined by the mean and standard deviation, the points whose average distance were outside the standard range (by the global distance mean plus the defined standard deviation) were defined as outliers and removed from the data. Then, the progressive triangulation densification filtering algorithm in TerraSolid software (2019, Finland, https://terrasolid.com/, accessed on 15 June 2020) was used to classify ground points and non-ground points [37]. The DEM generated by the initial automatic filtering results was visually checked to find irregular bumps and low surfaces. The corresponding point cloud with obvious misclassification was detected by the profile line, and the misclassification points were manually adjusted by the classification tool to improve the initial filtering results. DOM can be referred to determine the category of each point in the manual classification process. WGS-84 was adopted for all data coordinate systems to ensure the consistency of coordinate data.

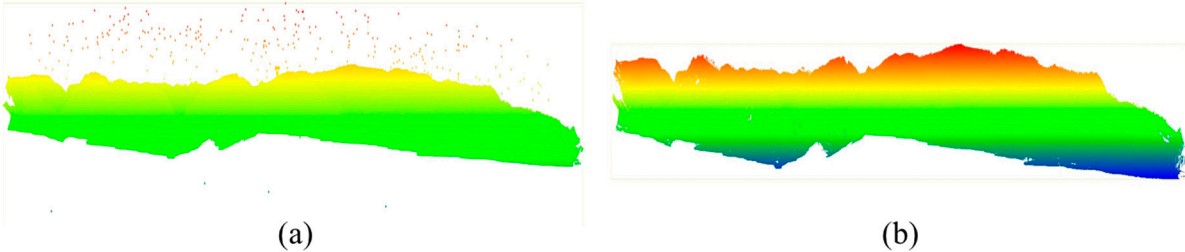

(a)　　　　　　　　　　　　　　　　　　　　(b)

**Figure 4.** Denoising process of point cloud. (**a**) Original point cloud; (**b**) Point cloud after noise removal.

## 3. Method

The study monitored the mining surface subsidence by integrating of the multi-temporal UAV photogrammetric data and airborne LiDAR data. The precision of sparse point cloud was evaluated through Monte Carlo simulation to illustrate the spatial variability of precision and the influencing factors of photogrammetry. Then, the airborne LiDAR point cloud was used for co-registration of the multi-temporal UAV dense matching point cloud to improve the consistency of the data. Finally, the M3C2 algorithm was applied to obtain the distance along the surface normal direction of the two point clouds, quantify the surface displacement, and reveal the spatio-temporal development characteristics of the dynamic moving basin. The technical flow chart of this study is shown in Figure 5.

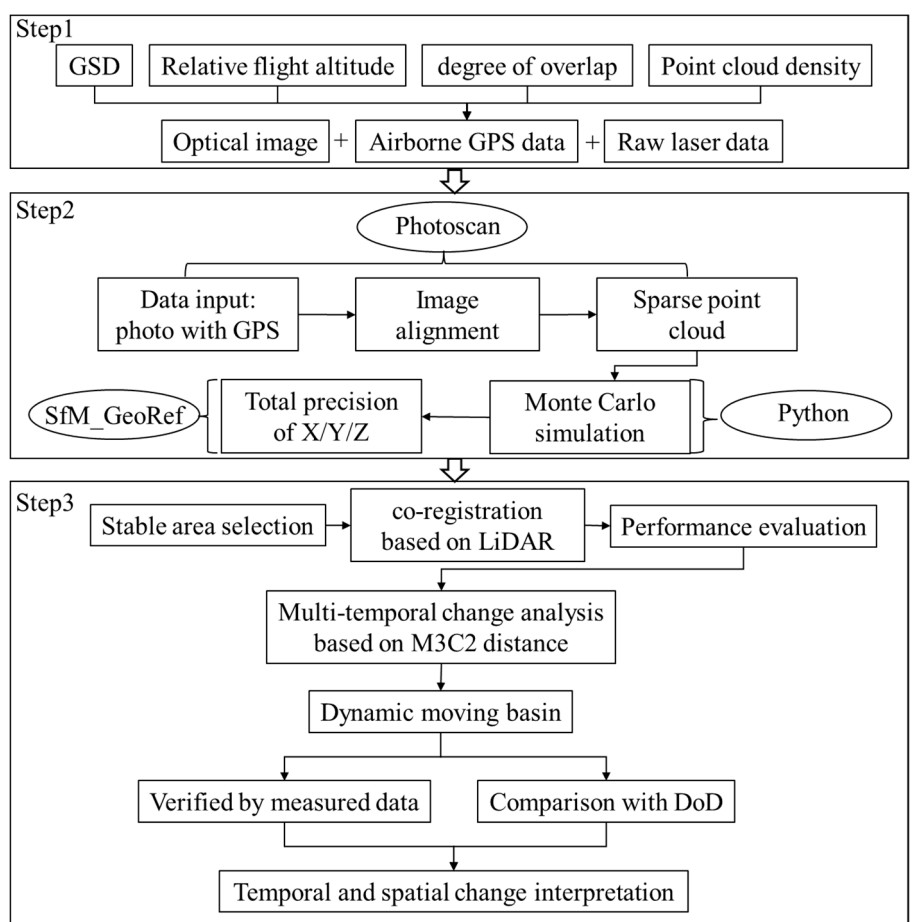

**Figure 5.** Technical flow chart of this paper.

### 3.1. Precision Estimation Based on Monte Carlo Simulation

Precision estimation is a part of strict photogrammetric processing. Here, precision refers to an expected standard deviation of the estimated value or measured value. The feature extraction and matching step are to identify the tie points (i.e., sparse point cloud)

in the image and match them in multiple images, and preliminarily estimate their three-dimensional point coordinates according to the observations of two-dimensional images. As shown in Figure 6, the position of tie points on the ground can be determined by observing the points in different images. Each tie point can be accompanied by a three-dimensional measurement precision ellipsoid [38]. When there is no georeferencing in Figure 6a, the uncertainty of the tie point leads to the uncertainty of the surface shape, as shown by the gray band around the black line. In Figure 6b, the correct orientation of the model in the ground coordinate system is determined by georeferencing. At this time, the uncertainty of the scale, translation and rotation of the model is small. For example, control measurement is carried out through the black ellipse representing GCP in Figure 6b.

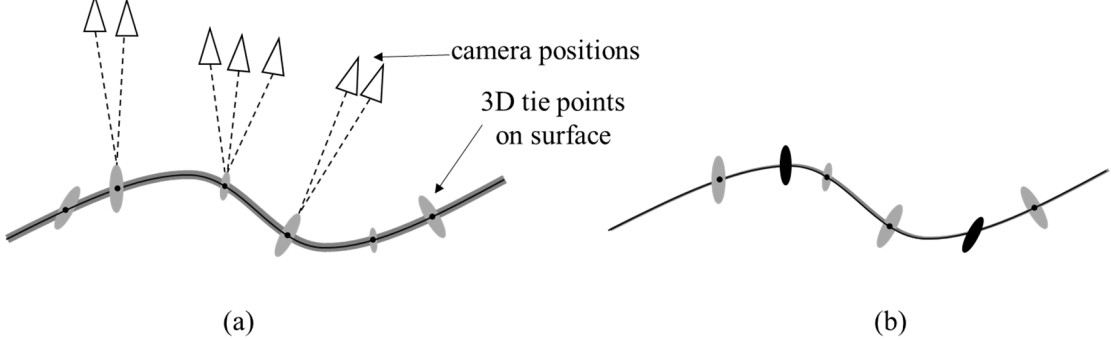

**Figure 6.** Schematic diagram of measurement precision and georeferencing. (**a**) No georeferencing; (**b**) With georeferencing. Adapted from [38].

To demonstrate how the uncertainty of the tie point changes spatially, the Monte Carlo method was used to generate the precision maps when using PhotoScan software. The precision map is a three-dimensional precision estimation of the tie point coordinates, and it can be used to define repeatability of measurements in the results, given the error associated with the input measurements (i.e., the tie point image observations). By using the Gaussian distribution iteration and randomly adjusting the camera model parameters, each point in the sparse point cloud will migrate around its original coordinates (i.e., the original position after the photo alignment operation) in 3D space, and the magnitude of point migration can be regarded as the proxy of product precision [38]. The image network was constructed by using the image alignment function to automatically match and orient images in PhotoScan. Subsequently, a Monte Carlo analysis was performed by calling the Python script in PhotoScan to automatically repeat the bundle adjustment. Finally, in the SfM_georef toolbox [11], all files output by the script iteration were compiled into point precision estimation, representing the variable precision of each point in the sparse point cloud in three different spatial dimensions. The precision maps reflect the factors affecting the point precision and the precision differences in the three directions of XYZ. Point precision is affected by a range of factors that are either internal to the photogrammetric network, such as imaging geometry and the quality of the tie point identification within the images, or related to the georeferencing. Precision maps showing broad, systematic variations indicate weakness in the overall survey georeferencing, which is symptomatic of weak control. If precision maps indicate strong localized variations, then photogrammetric factors are being expressed, e.g., differences in image measurement quality for individual tie points, and image network geometry aspects such as image overlap and convergence [38].

*3.2. Co-Registration of Multi-Temporal Point Clouds*

To improve the consistency of data in different periods, co-registration was carried out for the multi-temporal UAV dense matching point cloud. In vegetation covered areas, airborne LiDAR measurement has higher accuracy due to its penetration. Therefore, the airborne LiDAR point cloud of 16 January 2022 was taken as a reference. The iterative closest point (ICP) algorithm was used to conduct automatic co-registration for the dense

matching point clouds in June 2020, July 2020, September 2020, and July 2021, respectively. ICP was applied to a subset of point clouds located in stable areas that did not change between data acquisitions, and then the resulting transform parameters were applied to the complete point cloud to minimize the impact of georeferencing mismatches.

The surfaces near the control points arranged for the surface movement observation were taken as the stable area for the co-registration, and three of them were selected, as shown in Figure 7a. The control points were generally arranged outside the moving basin. The nearest distance between the three areas and the boundary of the working face is about 368 m, and the mining depth is 446.8 m, and the nearest distance accounts for more than 0.82 of the mining depth, the three areas are outside the area of mining influence and were not affected by mining. These areas have flat terrain, less vegetation coverage, and basically no subsidence, and the images obtained in these areas do not have the problem of insufficient overlap. Figure 7b represents the number of overlapping images used in the calculation of each pixel, while the red and yellow areas indicate a low degree of overlap and may produce poor results. The green area indicates that each pixel has more than 5 image overlaps, indicating that the data obtained from these stable areas will not be affected by insufficient image overlap.

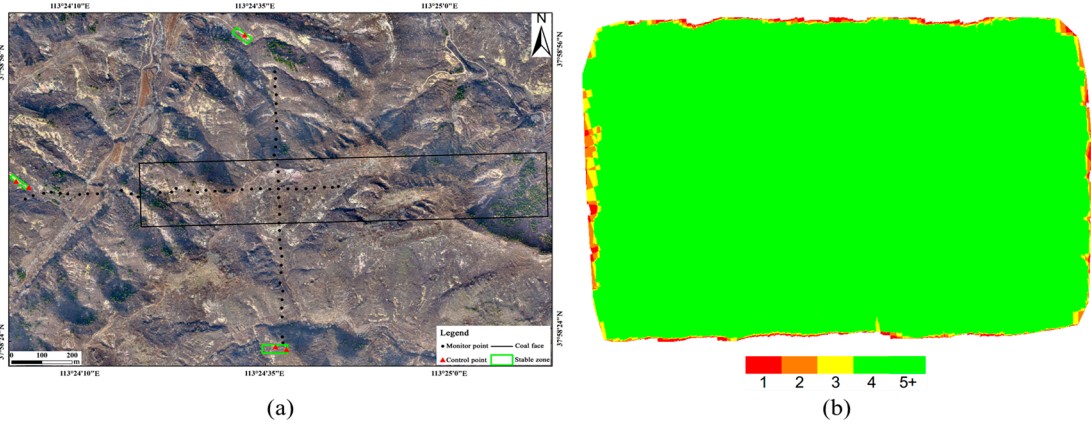

(a)                                   (b)

**Figure 7.** Selection of stable region: (**a**) Stable region selected by co-registration; (**b**) the number of overlapping images calculated for each pixel of DOM [39].

### 3.3. Quantification of Surface Change Based on Three-Dimensional Point Cloud

The M3C2 tool of the CloudCompare software [40] was used to calculate the distance between two adjacent photogrammetry point clouds. The distance between 06.14 and the last three photogrammetry point clouds was estimated to quantify the three-dimensional surface changes and reveal the spatial-temporal characteristics of surface deformation.

The M3C2 algorithm calculated the distance between the reference point cloud and the comparison point cloud relative to the local surface normal direction using user-defined normal scale D and projection scale d. Normal scale D was mainly used to create the best fit plane for the local neighborhood in the reference point cloud. The normal vector was used to project a cylinder of diameter d and calculate the average position of two clouds in the cylinder along the normal direction, and then give the local distance between the two point clouds. The uncertainty associated with each change was estimated by quantifying the level of detection (LoD). Using statistical *t* test and assuming that the error obeys normal distribution, LoD was calculated at 95 % confidence level [38]:

$$LoDetection = \pm 1.96(\sqrt{\frac{\sigma_1(d)^2}{n_1} + \frac{\sigma_2(d)^2}{n_2}} + reg) \quad (1)$$

where $n_1$ and $n_2$ are the number of local point clouds, $\sigma_1$ and $\sigma_2$ are the standard deviations of the orthogonal distance from the point to the best fit plane. In the M3C2 algorithm, these standard deviations are used as the measurement standard of point cloud roughness, and

reg is the registration error. Ultimately, only surface changes that exceeded the relevant LoD were considered statistically significant at a 95% confidence level.

## 4. Results

### *4.1. The 3D Precision of the Sparse Point Cloud*

For terrain change detection, it is necessary to quantify the precision of each surface. Figure 8 shows the tie point precision map measured at 06.14 through 1000 Monte Carlo iterations. The center of the study area has the greatest image overlaps, while the edge images do not have side overlap. The 3D point coordinate precision shows a correlation with the change in image overlap, i.e., the precision decreases as the distance from the center increases.

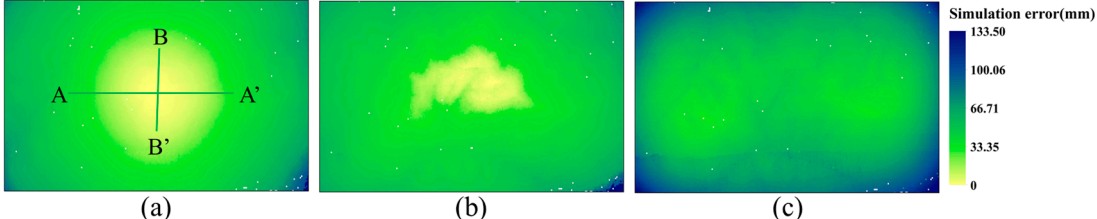

**Figure 8.** The 3D precision map of sparse point clouds based on Monte Carlo simulations. (**a**–**c**) represents the X, Y and Z directions, respectively.

Figure 9 shows the precision values along profile AA' and BB'. It was found that the precision was limited by the photogrammetry factors (i.e., image network geometry). The error propagation mode has different modes in X, Y and Z directions, respectively. The phenomenon that the center precision is not the highest is observed in the Z direction, which is called the dome effect, and the maximum error is still located at the edge of the study area. These precision changes can highlight areas with poor image coverage due to the partial occlusion in complex terrain or areas where image matching is difficult due to vegetation.

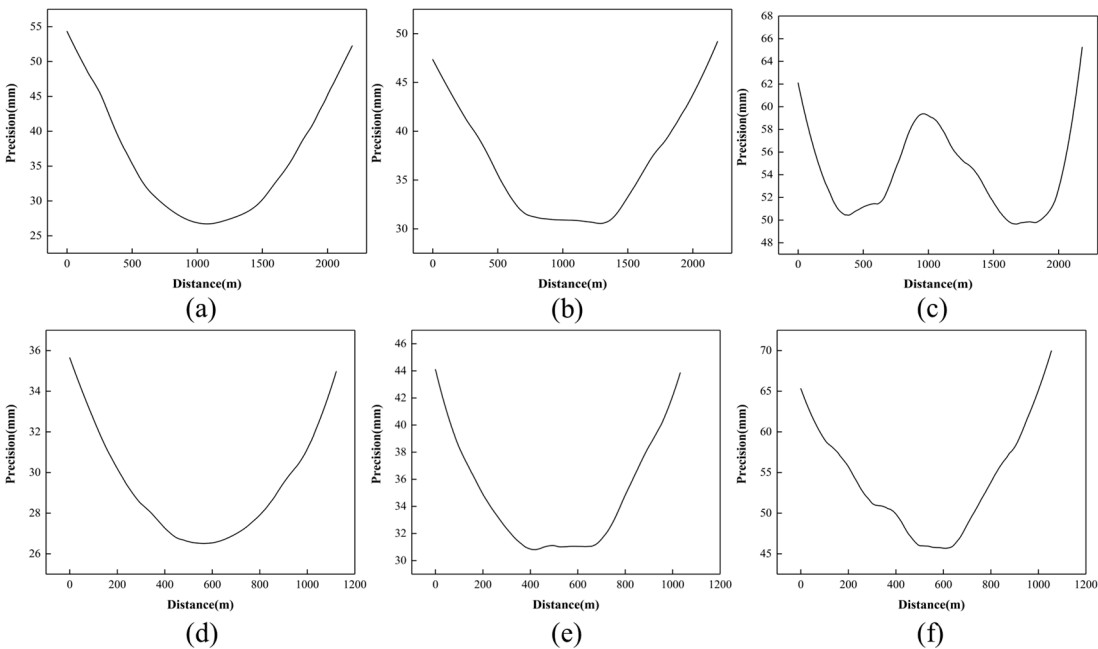

**Figure 9.** Precision values extracted from the profile lines in the XY and Z direction precision maps, respectively. (**a**–**c**) A-A' profile; (**d**–**f**) B-B' profile.

Figure 10 shows the three-dimensional precision histogram. Without clipping the result boundary, the precision error ranges are 0.026 m–0.093 m, 0.030–0.077 m, and 0.043–0.134 m for the X, Y and Z directions, respectively. The precision histograms in X and Z directions show positive skewness, and the Y direction is approximately normal distribution. The average precision in the Y direction is slightly larger than that in the X direction, but both are smaller than the Z direction, which is a foreseeable result because the elevation measurement is more affected by uncertainties.

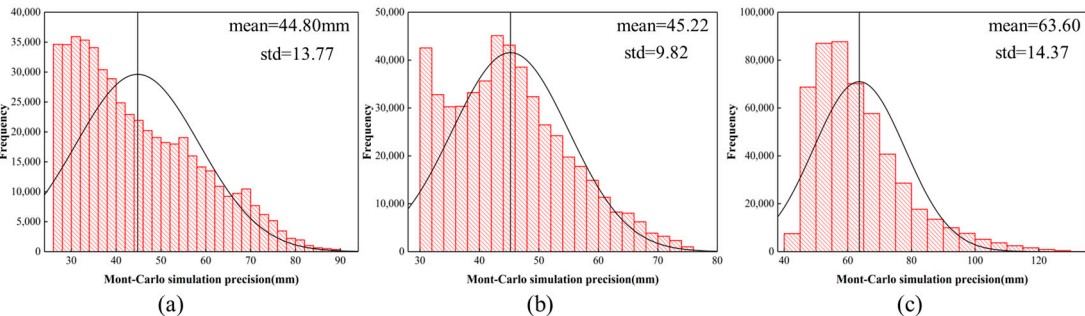

**Figure 10.** The 3D precision histogram of the sparse point cloud, based on Monte Carlo simulation, (**a**–**c**) represents the X, Y and Z directions, respectively.

### 4.2. Co-Registration Performance Evaluation

The basic statistics of the M3C2 distance are used as indicators of solution quality, where the standard deviation of the M3C2 distance is used as an indicator of measurement precision, and the average distance is considered as the accuracy of the point cloud. Table 3 shows the mean and standard deviation of the M3C2 distance in the stable region. Although the distance calculation was performed on the surfaces considered to be stable, the microtopography of these surfaces may change due to weathering or vegetation while maintaining the overall surface structure. Therefore, the result of co-registration may never reach the theoretical value of 0 m.

**Table 3.** The mean and standard deviation of M3C2 distance calculated in the stable region.

| Dataset | Mean (m) | Standard Deviation (m) | Duration (Day) | Platform |
|---|---|---|---|---|
| 01.16/06.14 | 0.24 | 0.13 | 581 | LiDAR/UAV |
| 01.16/07.20 | 0.34 | 0.15 | 545 | LiDAR/UAV |
| 01.16/09.07 | 0.30 | 0.19 | 496 | LiDAR/UAV |
| 01.16/07.31 | 0.35 | 0.15 | 169 | LiDAR/UAV |

Figure 11 shows the M3C2 distance histogram for each period. The distance distribution is approximately Gaussian distribution, and there is a certain degree of positive skewness. Among them, the mean of M3C2 distance calculated by 07.20 and 07.31 data is larger, which may be because of the dense vegetation at this time, which affected the results to some extent. In the study of multi-temporal monitoring of UAV, the repeatability (precision) of the data is more important than the accuracy of geolocation. The standard deviation of the M3C2 distance is between 0.13 and 0.19, indicating that the repeatability of the UAV multi-temporal photogrammetry data is comparable.

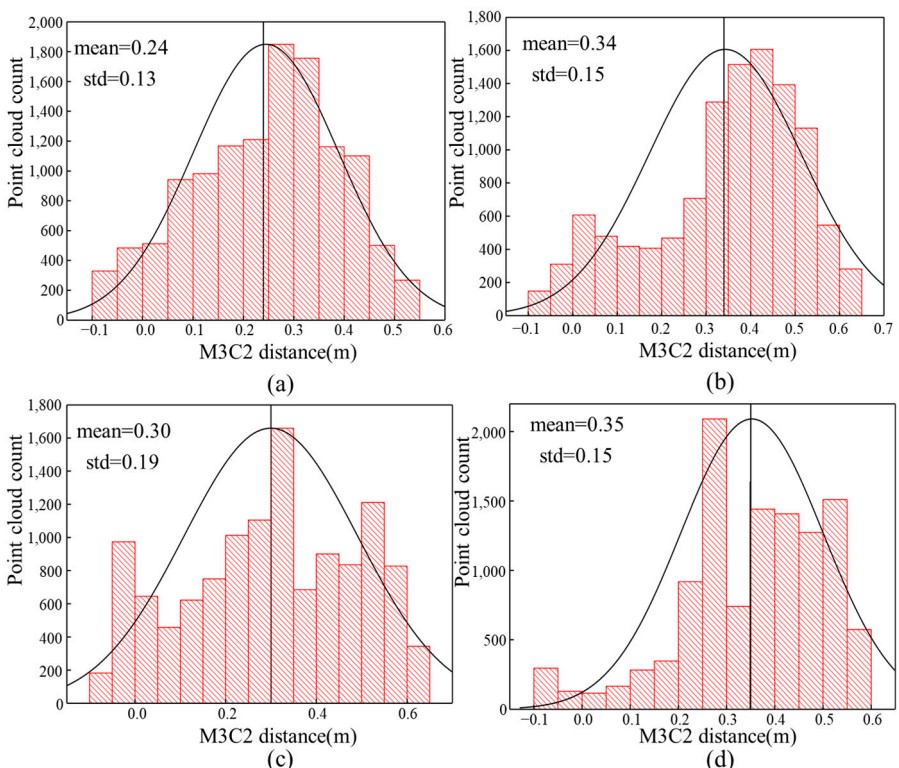

**Figure 11.** M3C2 distance histogram calculated in stable area. (**a**) LiDAR01.16-UAV06.14; (**b**) LiDAR01.16-UAV07.20; (**c**) LiDAR01.16-UAV09.07; (**d**) LiDAR01.16-UAV07.31.

### 4.3. Surface Change of Three-Dimensional Point Cloud

The multi-temporal three-dimensional topographic changes at different measured time intervals was obtained by calculating the M3C2 distance of the dense point cloud. Figure 12a–c shows the change maps obtained by subtracting the two adjacent point clouds and Figure 12d–f shows the cumulative change maps obtained by subtracting the point clouds of 07.20, 09.07, and 07.31 from the point clouds of 06.14, respectively. The red line segment is the advancing position of the working face at the corresponding time, which shows good consistency with the influence range of surface movement. It is observed that the surface in front of the working face is subsiding under the influence of mining. The surface moving basin is gradually formed during the advancement of the working face. With the advancement of the working face, the influence range of the surface expands accordingly, the maximum change value of the surface gradually increases, and the maximum deformation range moves closer to the center of the goaf. Abnormal values with large deformation were observed outside the working face, which may be due to the error caused by the low density of ground points after filtering due to dense vegetation. The reason for the greater values in Figure 12a may be that the vegetation was dense at this time, blocking the ground and affecting the acquisition of topographic data.

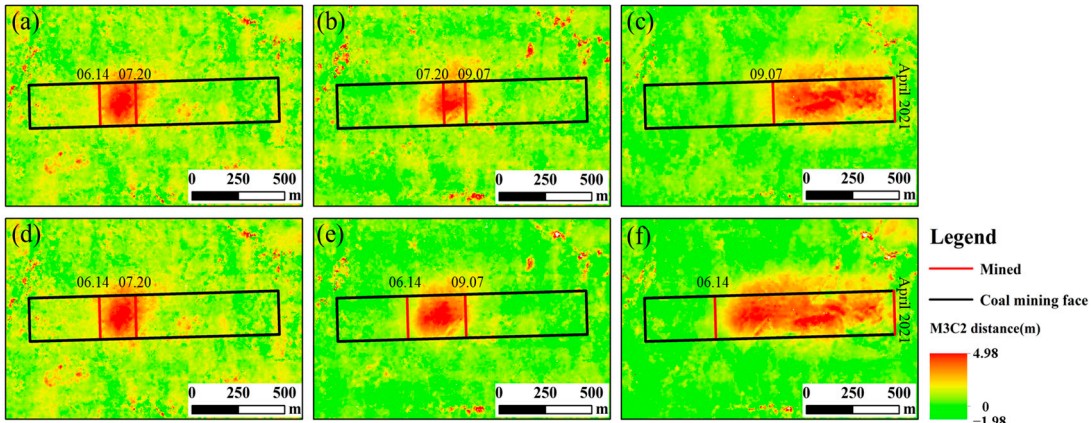

**Figure 12.** The mining-induced subsidence during 2020.06.14–2021.07.31. (**a**) 07.20–06.14; (**b**) 09.07–07.20; (**c**) 07.31–09.07; (**d**) 07.20–06.14; (**e**) 09.07–06.14; (**f**) 07.31–06.14.

The positive value of the calculated M3C2 value represents the ground subsidence. Obvious surface changes can also be seen from the statistical histogram of cumulative changes in Figure 13, and the mean and standard deviation of the distance increase with the increase of the time interval. With the advance of the working face, more and more surface points are subsiding under the influence of mining.

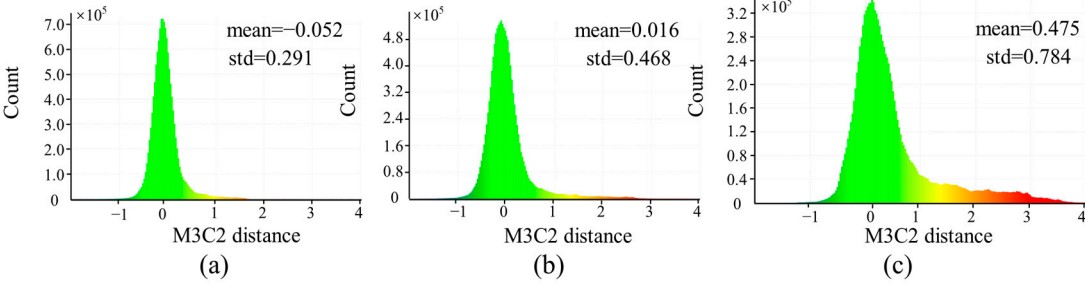

**Figure 13.** Histogram of M3C2 value distribution calculated by multi-temporal dense point cloud. (**a**) 07.20–06.14; (**b**) 09.07–06.14; (**c**) 07.31–06.14.

The surface changes calculated by the M3C2 distance were compared with the field monitoring points data measured by the total station/GPS RTK and the difference of the DEM (DoD) method to quantitatively verify the accuracy of the M3C2 distance. The range of 1 m$^2$ is determined by taking each measured monitoring point as the center, and all point clouds in the range are searched to calculate the average of the M3C2 distance. For each monitoring point, the average of the M3C2 distance value was used as a proxy and compared with measured data. Taking the data of 06.14–07.20 and 06.14–07.31 as examples, the mean error (ME), mean absolute error (MAE) and root mean square error (RMSE) were calculated simultaneously. The calculation results are shown in Table 4, where the mean error is small, basically in the range of ±0.2 m. The average error of the M3C2 distance in the 07.31–06.14 result of line A is greater than the DoD, which may be caused by the offset of positive and negative errors caused by a large error value. The average absolute error is the average of the absolute values of the deviations between all individual observations and predicted values, which can better reflect the actual error. In addition, because it is proportional to the square of the deviation, the RMSE is greatly affected by the outlier error, generally greater than the ME and the MAE, mostly between 0.2 m and 0.3 m. Due to the influence of time decoherence, the data with the larger time interval with 06.14 has greater RMSE. Compared with the DoD extraction values, the M3C2 distance values show the lower RMSE in both line A and line B.

**Table 4.** Comparison error calculation between measured points and subsidence value extracted by UAV.

| Dataset | Mean Error (m) | | Mean Absolute Error (m) | | Root Mean Square Error (m) | |
| --- | --- | --- | --- | --- | --- | --- |
| | Line A | Line B | Line A | Line B | Line A | Line B |
| DoD 07.20–06.14 | −0.16 | −0.11 | −0.20 | 0.14 | 0.24 | 0.17 |
| M3C2 07.20–06.14 | −0.14 | −0.10 | 0.18 | 0.14 | 0.22 | 0.16 |
| DoD 07.31–06.14 | 0.06 | −0.28 | 0.31 | 0.29 | 0.34 | 0.32 |
| M3C2 07.31–06.14 | −0.13 | −0.14 | 0.19 | 0.23 | 0.23 | 0.25 |

Taking the intersection of line A and B as the origin, the comparison among the measured deformation, the M3C2 distance and the DoD subsidence value is shown in Figure 14. The two curves extracted from the UAV data are close to the trend of the measured subsidence curve. The deviation is small near the measured maximum subsidence value, and the M3C2 distance value alleviates the problem of more elevation jump points in the UAV data compared with the DoD extraction value. The difference between field measured data and UAV measurements may be caused by the following factors: (1) the time between the UAV and the GPS monitoring point measurement is not completely matched; (2) The spatial position between the UAV extraction point and GPS monitoring station is inconsistent.

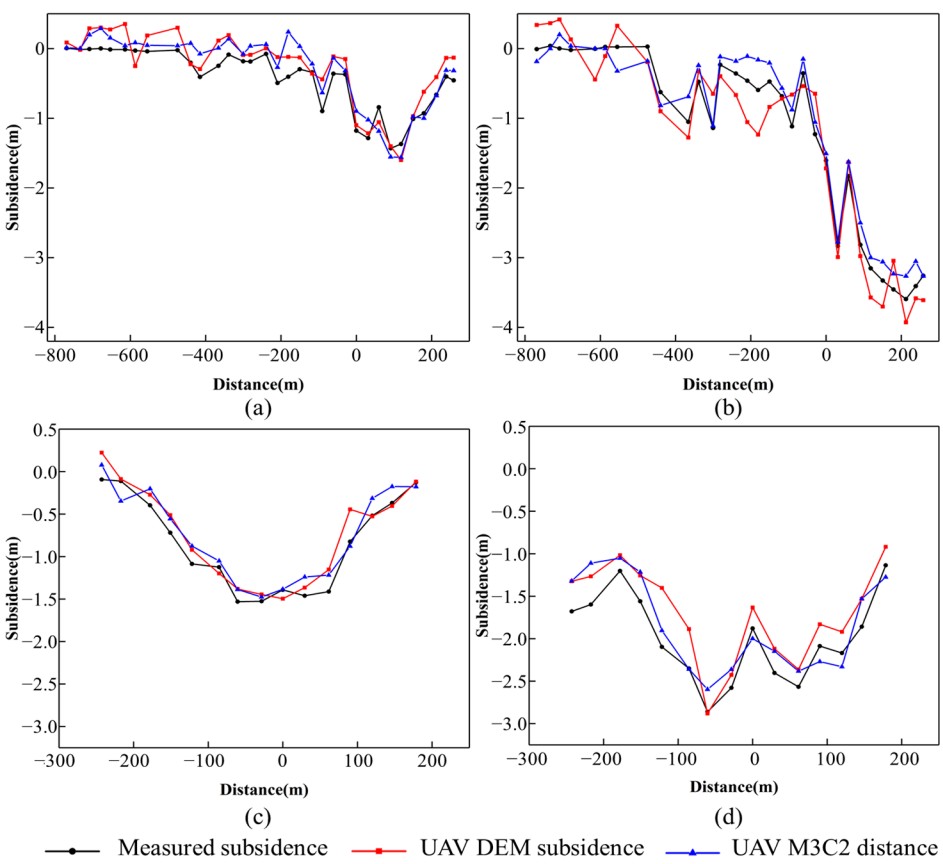

**Figure 14.** Comparison among the measured subsidence, M3C2 distance values and DoD extraction subsidence at monitoring points. (**a**) Line A 07.20–06.14; (**b**) Line A 07.31–06.14; (**c**) Line B 07.20–06.14; (**d**) Line B 07.31–06.14.

## 5. Discussion

### 5.1. Precision Analysis of Monte Carlo Simulation

In this study, the mean value of precision error in the Y direction is slightly larger than that of the X direction, and the cross-flight can be used to reduce the difference in precision

between X and Y directions. The dome effect is observed in the Z direction precision map, which is also recorded in the evaluation of the UAV SfM photogrammetry by Rosnell and Javernick et al. [41,42]. Parallel flight will lead to a dome effect in the vertical direction [43]. The dome effect can be alleviated by strengthening the network geometry by including oblique images [11].

According to the study of Carbonneau et al. [44], the error of 0.06 shows that the optimal precision of flight height is 0.1% when the flight height is 60 m, which means that 1:1000 can be regarded as the inherent limit of the SfM precision level. The median ratio between the RMSE and survey range is about 1:640 in more than 40 published studies [45]. Our maximum error in the Z direction is 0.134 m and the ratio of precision to observation distance is about 1:1900 at the relative ground flight height of 255 m, which is similar to the results of Li et al. in the polar region, where the maximum precision of sparse point clouds is 0.83 m and the ratio is about 1:1800 at the flight height of approximately 1500 m [46]. Therefore, we can classify our terrain data as high-quality data. In terms of image ground sampling distance, the simulated horizontal mean is roughly the same as that of the GSD (4 cm), and the vertical mean precision is about 1–3.5 times that of the GSD, which has similar results in previous studies [46,47].

Regarding the number of iterations, the Monte Carlo simulation processing in this manuscript took several hours, and the precision of simulation results was acceptable. Considering the tradeoff between processing efficiency, storage space and result precision, we set the number of iterations to 1000, which is the same as the research of Zhang et al. [32].

### 5.2. Advantages of Monte Carlo Simulation

Calculating the RMSE value at the ground checkpoint is a commonly used error evaluation method when the ground data is a set of distribution points rather than a continuous surface. This operation does not fully describe the spatial variable precision of SfM photogrammetry and the lower average error value may mask the larger uncertainty [44]. Error assessments thus benefits from a larger number of more evenly distributed checkpoints, which are not easy to carry out in some remote areas. The three-dimensional precision of the UAV SfM sparse-point-cloud-based Monte Carlo simulation is an effective method to analyze the uncertainty of photogrammetry and direct georeferencing.

The Monte Carlo iterations generate not only the precision map but also the associated covariance. Therefore, the point coordinate error ellipsoid can be derived and the correlation between the camera parameters can be evaluated, which is helpful to check the over-parameterization of the camera model [38]. The layout of ground control points is inevitable when there is a high requirement for the quality of the results. Meanwhile, it is still necessary to achieve a compromise between the measurement time and the quality of surface reconstruction. The different numbers of GCPs can be used to evaluate the change of precision at GCPs and checkpoints within different GCP density ranges, and to estimate the minimum GCP density that meets specific precision requirements, minimizing the relevant field work with minimal impact on measurement precision [38,48].

Attention should be paid to error control in the UAV SfM processing process, but most software cannot provide detailed precision information at present. As emphasized by Micheletti et al., SfM has the advantage of providing a black box tool without expert supervision, and may become a disadvantage, resulting in less user participation in data quality control, and the source of data error may not be identified [49]. Using code to flexibly operate the UAV SfM project is helpful to accurately understand the error size, spatial variability and error sources, so as to improve the accuracy of results based on UAV SfM [50].

### 5.3. Methods to Improve the Quality of UAV SfM Results

Many recent studies have shown that there are systematic errors in the processing flow of automated UAV SfM [44,51]. This systematic error usually comes from image sensor characteristics, camera distortion model, imaging network geometry, surface texture,

weather conditions, and camera parameter over-parameterization. It is time-consuming and expensive to establish a well-distributed control point network, and it is also dangerous in mountainous areas. Therefore, some measures can be taken to improve the quality of UAV SfM results using direct georeferencing: additional UAV cross-flight routes, adjustment of flight speed to reduce the risk of image blur at high speed, and the acquisition of images with large overlap, strengthens the geometry of the imaging network, and improves the reliability of image matching; adding additional oblique images to the network to provide convergent image acquisition [52]; selecting a period of time with clean air and sufficient light for data acquisition; and a well-defined camera distortion model [50].

### 5.4. The Advantages of M3C2 Compared with DoD

For complex terrain scenarios, M3C2 is described as the most suitable point cloud quality descriptor [53]. The M3C2 algorithm runs directly on the point cloud without grid division and avoids the errors introduced in the process of point cloud interpolation to DEM, so it is widely used in the research of point cloud change detection [7,54]. The M3C2-based method is more suitable for detecting changes in complex three-dimensional environments than DoD [40]. DoD may overestimate errors on steep terrain while small lateral shifts may produce large vertical differences. Figure 15 shows the comparison between M3C2 and DoD during 06.14–07.31. It is clear that M3C2 algorithm is smoother than the DoD method, where significantly more noise in the form of white grids is observed. The M3C2 method qualitatively shows the improvement in the reliability of surface deformation distribution.

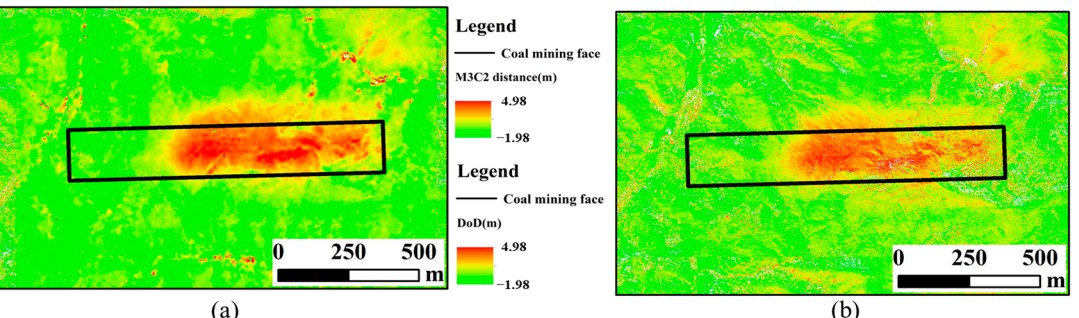

**Figure 15.** Comparison between M3C2 and DoD during 06.14–07.31. (**a**) M3C2 distance; (**b**) DoD.

The multi-temporal LiDAR data in this study is expected to achieve better results, because the penetration of LiDAR makes it possible to obtain the terrain under vegetation. In addition, the long-period surface change sequence can be combined with the vegetation change data to reduce the influence of vegetation on multi-temporal monitoring of UAV by analyzing the change law of surface vegetation cover.

The spatial and temporal resolution of data collection should be adapted to the dynamic changes observed in the scene, and the appropriate monitoring frequency depends on the stage of the surface movement process. In the active stage of surface point movement, the subsidence rate is greater than 5 cm/month, and the subsidence is mainly concentrated in this stage. At this time, increasing the number of observations can better reveal the law of surface movement caused by mining and avoid cumulative effects.

Due to the influence of local terrain roughness and point cloud density, the calculated M3C2 distance detection level threshold limits the application of UAV to obtain small-scale data such as horizontal surface movement. With the advancement of UAV hardware technology and the development of computer vision technology, multi-temporal point clouds based on high density and high precision have the opportunity to reveal finer surface deformation.

## 6. Conclusions

In this paper, multi-temporal data collected by UAV with visible camera and LiDAR sensor were used to monitor mining-induced subsidence. The main conclusions can be summarized as follows:

(1) Monte Carlo simulation is an effective method to evaluate the relative precision of photogrammetric products without ground checkpoint. Repeated bundle adjustment through Monte Carlo simulation iteration can generate 3D precision maps of sparse point clouds. The results show that the average precision in the three directions is 44.80 mm, 45.22 and 63.60 mm, respectively. This operation is helpful to understand the factors affecting the precision and optimize the future measurement planning.

(2) Co-registration of multi-temporal UAV data can reduce dependence on GCPs. The consistency between different measurements is very important when comparing multi-temporal measurement data. Taking the standard deviation of the M3C2 distance as the index, the repeatability of multi-temporal UAV SfM data was evaluated by airborne LiDAR data. The results show that the standard deviation of the M3C2 distance is between 0.13 and 0.19, indicating the comparable results of UAV multi-temporal photogrammetry data.

(3) The surface displacement related to mining activities along the local normal direction among multi-temporal three-dimensional point clouds was obtained from the M3C2 algorithm. The results show that the M3C2 algorithm based on three-dimensional point clouds can obtain subsidence information and identify the characteristics of dynamic moving basin development. It is a valuable supplement to the traditional 2.5D method for analyzing topographic changes, and can provide reference for the monitoring of similar objects such as landslides and rock glaciers.

**Author Contributions:** Conceptualization, X.L. (Xiaoyu Liu) and X.L. (Xugang Lian); methodology, X.L. (Xiaoyu Liu) and W.Z.; software, X.L. (Xiaoyu Liu); validation, W.Z.; formal analysis, X.X.; investigation, X.L. (Xiaoyu Liu) and X.L. (Xugang Lian); resources, X.L. (Xugang Lian); data curation, W.Z.; writing—original draft preparation, X.L. (Xiaoyu Liu); writing—review and editing, W.Z.; visualization, X.X.; supervision, W.Z.; project administration, X.L. (Xugang Lian); funding acquisition, W.Z. All authors have read and agreed to the published version of the manuscript.

**Funding:** This research was funded by Chang'an University (Xi'an, China) through the Natural Science Foundation of China (NSFC) (Nos. 42074040, 41941019), the National Key R&D Program of China (2020YFC1512001), Fundamental Research Funds for the Central Universities, CHD (No: 300102262902) and Natural Science Basic Research Plan in Shaanxi Province of China (2023-JC-JQ-24).

**Institutional Review Board Statement:** Not applicable.

**Informed Consent Statement:** Not applicable.

**Data Availability Statement:** Not applicable.

**Acknowledgments:** We would like to thank the reviewers for their helpful comments.

**Conflicts of Interest:** The authors declare no conflict of interest.

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
