# Peer review of "Monitoring Mining Surface Subsidence with Multi-Temporal Three-Dimensional Unmanned Aerial Vehicle Point Cloud"

_remotesensing, doi:10.3390/rs15020374_

Round 1

Reviewer 1 Report

It is important to observe the mines with up-to-date technological tools such as UAV and LIDAR. The study is interesting, but there are some shortcomings. I suggest correcting them so that the reader can better understand them.

1.       The abbreviation of HRT (high-resolution terrain) is not a common usage. Already mentioned once in the text. no need for abbreviation.

2.       Check the section numbering. Some third-degree titles are wrong ( such as  2.1.1-2.1.2).

3.       Since the LiDAR penetrate to the vegetation and reach the terrain UAV datasets must filter to get ground points. Otherwise M3C2 distance comparison becomes inconsistent.

4.       The explanation of DEM abbreviation should be given because it is not clear whether surface model or terrain model was used in the study.

5.       It should be clearly stated what effect the Monte Carlo analysis has on the data used.

6.       If the RMSE values are around 20 cm (as author said and given in table 4), how reliable are the 6 cm surface subsidence.

7.       Please specify dates in figure 15 Captions

8.       The datasets are very dense. For this reason, it will be useful for the readers to give information about the hardware used and the processing times.

Reviewer 2 Report

The article presents the use of the M3S2 algorithm for the analysis and determination of vertical displacements of the ground surface under the influence of underground hard coal mining. It raises the current topic of monitoring mining trains with the use of UAV and LIDAR. A difficult mountainous terrain was chosen as the testing ground. The presented results indicate the high usefulness of the discussed method, and the results are characterized by greater accuracy than the DoD for DEM used so far.

In my opinion, the article is at a high level, but it needs some additions and corrections.

Below are my comments on the article.

Do the subsidence maps obtained with the M3C2 method take into account the horizontal components of displacements. The same problem applies to the leveling of observation points in mining areas.

Chapter 2.1 - When the exploitation of the longwall was completed?

Table 1 - Please correct header of last column.

Figure 7 - Please refer to the previous article from which the graphic is taken https://doi.org/10.3390/app12189374

Line 292 - Is 1000 iteration of Monte Carlo simulation sufficient to analysis?

Table 3 – The average values in this table indicate vertical displacements in the stable region. This is a contradiction to the line 263-264 (‘and basically no subsidence’). Please explain this.

Lines 333-340 - How to interpret the results obtained from MC Simulation in relation to the standard deviation in Table 3. Can MC Simulation results be defined as internal accuracy?

Figure 12 - Please add the UAV measurement date in Figures c) and f) above the red lines (‘Mined’ in the legend). Please also correct the description of Fig. 12 e) and f) and adding the value 0 in the color scale.

Figures 12 and 13 - Please unify the color scale in the legend in Figure 12 and the colors in the histograms in Figure 13.

Line 375 - Please add information about which points were measured using GNSS technology. What was accuracy of XYZ coordinates? Lines 156-164 only mention Total Station measurements.

Line 378 – Please use mean error instead of average error.

Lines 380-381 - in relation to this fragment, please provide information on the accuracy of the DEM height (in each series). Large error value should be visible in the histogram. What about outliers – did you use any methods to remove them?

Line 397 - Have you used your own base station (or a GNSS receiver as a base in a static GNSS survey) or public data from a permanent GNSS station? For the latter, data on the stability and accuracy of the XYZ coordinates should be available.

Figure 15 - Please add one more map in Figure 15 as difference a) - b). This will give a better picture of comparing both methods of determining land surface subsidence.

Line 488 – Analyzed exploitation shows in Figure 12a much more greater values

Round 2

Reviewer 1 Report

Thank you to the authors for considering the suggestions.

Author Response

Thank you again for your constructive suggestions and insightful comments.

Reviewer 2 Report

Thank you for your extensive responses to questions and comments.
Last note: you only used GPS signal or others - in such case, please change GPS on GNSS.
